# The Dysfunction of Ca^2+^ Channels in Hereditary and Chronic Human Heart Diseases and Experimental Animal Models

**DOI:** 10.3390/ijms242115682

**Published:** 2023-10-27

**Authors:** Irina Shemarova

**Affiliations:** I.M. Sechenov Institute of Evolutionary Physiology and Biochemistry of the Russian Academy of Sciences, 194223 Saint-Petersburg, Russia; irina-shemarova@yandex.ru

**Keywords:** animal model, cardiac arrhythmias, cardiac calcium channels, calcium channelopathies, gene regulation, HCN channels, heart diseases, knockout model, RyR2, TRP channels

## Abstract

Chronic heart diseases, such as coronary heart disease, heart failure, secondary arterial hypertension, and dilated and hypertrophic cardiomyopathies, are widespread and have a fairly high incidence of mortality and disability. Most of these diseases are characterized by cardiac arrhythmias, conduction, and contractility disorders. Additionally, interruption of the electrical activity of the heart, the appearance of extensive ectopic foci, and heart failure are all symptoms of a number of severe hereditary diseases. The molecular mechanisms leading to the development of heart diseases are associated with impaired permeability and excitability of cell membranes and are mainly caused by the dysfunction of cardiac Ca^2+^ channels. Over the past 50 years, more than 100 varieties of ion channels have been found in the cardiovascular cells. The relationship between the activity of these channels and cardiac pathology, as well as the general cellular biological function, has been intensively studied on several cell types and experimental animal models in vivo and in situ. In this review, I discuss the origin of genetic Ca^2+^ channelopathies of L- and T-type voltage-gated calcium channels in humans and the role of the non-genetic dysfunctions of Ca^2+^ channels of various types: L-, R-, and T-type voltage-gated calcium channels, RyR2, including Ca^2+^ permeable nonselective cation hyperpolarization-activated cyclic nucleotide-gated (HCN), and transient receptor potential (TRP) channels, in the development of cardiac pathology in humans, as well as various aspects of promising experimental studies of the dysfunctions of these channels performed on animal models or in vitro.

## 1. Introduction

Calcium ions are vital for the normal functioning of the heart. They support the myocardial contraction of the heart and play a major role in regulating excitation–contraction coupling. Calcium current impacts the pacemaker activity and action potential formation (plateau phase) in cardiomyocytes. In addition, calcium ions are involved in cellular signaling, carrying out other general biological cellular functions, such as gene transcription, cell growth, and cell remodeling [1].

Cardiac Ca^2+^ channels are classified by their calcium selectivity and conductance as well as structure, the kinetics of activation and inactivation, and pharmacological characteristics [2].

The condition of these channels is regulated by (1) changes in the membrane potential effecting their gate mechanism (voltage-gated channels, VGCCs) [3]; (2) depletion of Ca^2+^ stores (store-operated Ca^2+^ channels) [4]; (3) Ca^2+^ or other intracellular stimuli (transient receptor potential (TRP) channels) [5]; (4) hyperpolarization and cyclic nucleotides (Ca^2+^-permeable nonselective cation HCN channels) [6]; and some others [7,8]. The predominant Ca^2+^ channels in the heart are highly selective voltage-gated channels, Cav channels (L- and T-types); however, an additional contribution to the provision of heart functions is made by nonselective ion channels that pass Ca^2+^ into cardiomyocytes (transient receptor potential (TRP) channels and hyperpolarization and cyclic nucleotide (HCN) channels [1,3,4,5,6,8].

Impairment of the physiological functions of Ca^2+^ channels leads to mitochondrial Ca^2+^ overload, damage to CM, apoptosis, and heart pathology. Currently, significant progress has been made in understanding the function of Ca^2+^ channels in a healthy heart, but there is still a limited amount of data on cytoarchitectonics and the physiology of Ca^2+^ channels in the cardiovascular cells associated with heart diseases. The dysfunction of voltage-gated calcium channels that occurs during pathophysiological processes in the heart leads to arrhythmias caused by early afterdepolarization (EAD) [9]. It has been shown in experimental animal models and in vitro that an overload of cardiomyocytes with calcium ions caused by the dysfunction of nonselective calcium channels resulted in delayed afterdepolarization (DAD) and irregular conduction cycles [10].

The problems of the origin and treatment of congenital and acquired Ca^2+^ channelopathies, the influence of Ca^2+^ channel mutations on the pathogenesis of chronic heart diseases, and associated multisystem disorders are also far from being fully understood. In addition, the contribution of posttranslational modifications of Ca^2+^ channels and the defects of Ca^2+^ handling proteins to the development of potentially lethal heart diseases are still poorly understood [11,12].

The purpose of this review is to highlight the little-studied aspects of the effect of Ca^2+^ channel dysfunction on the development of chronic and hereditary heart diseases.

## 2. Cardiac Voltage-Gated Calcium Channels (VGCCs)

The voltage-gated calcium channels (Cav channels) are an integral element of the sarcolemma of cardiomyocytes (CMs). Unlike other transport systems of cell membranes (carriers, pumps), they are able to pass Ca^2+^ at a rate several orders of magnitude higher in the open state. This determines their dominant role in ensuring the basic functions of the heart (contractility and pacemaking). The influx of Ca^2+^ contributes to keeping the membrane potential more positive, and the released calcium from the sarcoplasmic reticulum induces heart contraction [2]. This is a well-known phenomenon called “calcium-induced calcium release”. As a result, the excitation–contraction coupling process in cardiac myocytes can proceed without interruption [1].

Cardiomyocytes contain three types (L-, R-, and T-) of voltage-gated calcium channels that activate upon membrane depolarization. They differ in their properties, functions, and distribution in different compartments of the heart. Most importantly, these channels differ in the activation threshold, which is essential for ensuring the cellular specialization of cardiomyocytes [1,3,6,13,14].

### 2.1. L-Type Cav Channels (Cav1)

The main cardiac Cav channels are L-type Cav channels (LTCC—long-lasting large-capacity). These are high-threshold (HVA) ion channels, the activation threshold of which is significantly higher than in low-voltage-activated T-type calcium channels (LVA) [15]. It should be noted that Cav1 channels have electrophysiological properties that are characterized by high conductivity and very slow inactivation kinetics [2,15,16]. These properties of the channels ensure the generation of the action potential (in the plateau phase), the maintenance of normal sinus rhythm, and the excitation–contraction coupling (ECC) of cardiac cells in the depolarization phase of the action potential (AP) [17]. The L-type Cav channels (Cav1.2 and Cav1.3) are mainly localized in contractile cardiomyocytes and pacemaker cells of the cardiac conduction system [1,2,3,17,18]. It has been established that even minor changes in the function of calcium channels disrupt the plateau phase of the cardiac action potential and lead to an irregular conduction cycle [17], arrhythmias, and heart disease-associated pathophysiological conditions [1].

Cav1 channels are heterooligomeric protein complexes consisting of five subunits: α1, α2, δ, β, and γ. The α1 pore-forming subunit is the main component of the channel, and the remaining four subunits are auxiliary components (Figure 1).

However, the functioning of the channel and its positioning on the membrane requires the participation of all protein subunits [16]. It is noted that expression of the α1 subunit is sufficient to produce functional Ca^2+^ channels (in the example of skeletal muscles), but it has a low expression level and abnormal kinetics, and its voltage is dependent in the Ca^2+^ current [19]. Moreover, a few other effectors and regulatory proteins directly associated with α1-subunit are required to regulate Ca^2+^ transport and gating. These proteins (small and large GTPases, calmodulin, etc.) form supramolecular signal complexes with the α1-subunit of the Cav1 channel and significantly expand the repertoire of mechanisms that regulate the Ca^2+^ channel influx (Figure 2).

Previously, in animal models and in vitro, the mechanisms of the stimulating effect of catecholamines on myocardial contractility were studied, and it was found that the activation of β-adrenergic receptors increases L-type Ca^2+^ currents through PKA-mediated phosphorylation of the Ca_v_1.2 channel protein and/or associated proteins [20]. To date, a lot of data have accumulated, indicating the important role of these signal complexes in physiological processes and cardiac pathophysiological conditions [1,3].

Voltage-gated calcium channels LTCC exist in three, Cav1.1, Ca1.2, and Cav1.3 (Cav1.1–1.3), isoforms, two of which, Cav1.2 and Cav1.3, are found in CM and slightly differ in the structure of α-subunits encoded by the *CACNA1C* and -*D* genes, respectively [16,21]. Cav 1.2 and Cav 1.3 channels possess high conductivity (25 pSm), very slow inactivation kinetics (τ > 500 ms), and the ability to activate at high membrane potentials (over −10 mV).

In adult cardiac myocytes, calcium flows through the Cav1.2 channel and forms the main type of internal current during the plateau phase of the cardiac AP, and Cav1.2 is the dominant channel involved in ECC. Calcium currents also influence the electrical properties of cardiomyocytes. Cardiomyocytes’ channel dysfunctions affecting the AP profile are associated with various cardiac arrhythmias. Studies conducted using a canine ventricular cell model have shown that the inactivation of Cav1 channels, including Ca-dependent inactivation, is associated with acute Ca^2+^/calmodulin-dependent protein kinase II (CaMKII) overexpression and calcium handling dysfunction [22].

Ventricular myocytes have only Cav1.2 channels, whereas both Cav1.2 and Cav1.3 channels are expressed in atrial myocytes, the Gis–Purkinje conducting system, SA and AV node pacemaker cells, and the smooth muscle cells of blood vessels. Nowadays, alternative LTCC splicing has attracted attention as an instrument of tissue specificity, which revealed that the dominant variant of the Cav1.2 channel in smooth muscle cells differs from the one in heart cells [2,21]. The activation threshold of Cav1.2 channels is less negative (−30 mV) than in Cav1.3 channels (−50 mV), which is essential to ensure the sequence of electrical activity in the myocardium [12,18]. Failure of the throughput function of Ca channels (increase or decrease in the time of their open state, even without a genetic change in their structure) can lead to serious impairment of the heart function. Under some conditions, the dysfunction of L-type calcium channels will cause early afterdepolarization, a common aspect of cardiac arrhythmias [9,10].

Under some conditions, the dysfunction of L-type calcium channels will cause early afterdepolarization (EAD), which arises on the shoulder of a preceding action potential plateau and is favored by a slow preceding activation rate and prolonged action potentials. In this case, Cav1.2 channels are responsible for the inward current for EADs. These afterdepolarizations have characteristics that suggest their etiological role in cardiac arrhythmias found in heart failure [23].

The role of Cav1.3 channels is not limited only to their participation in the generation of diastolic depolarization in pacemaker cells. The opening of these channels contributes to the generation of local diastolic intracellular Ca^2+^ releases (LCRs) and is required for the coupled-clock system that drives the automaticity of human sinoatrial nodal pacemaker cells [24]. Unsurprisingly, malfunction of these channels leads to SAN dysfunction, atrioventricular conduction disorders, arrhythmias, and heart failure [17,18].

Mutations found in gene-encoding Cav1 channels determine a wide range of diseases called calcium channelopathies, and all four gene-encoding α- and β-subunits carry such mutations [17]. Cav1-channelopathies include muscular, neurological, cardiac, and visual syndromes. Among them is Timothy syndrome, which is manifested by prolongation of the QT interval and congenital heart defects [25]. This condition is associated with a high risk of sudden cardiac death (SCD) and is caused by defects in the *CACNA1C* gene encoding in the α1-subunit of the Cav1.2 channel [25,26]. Mutations in the *CACNA1C* gene change the structure of α-subunits and the conformation of Cav1.2 channels. As a result of these mutations, these channels remain open longer than usual, which leads to an excessive intake of Ca^2+^ into the heart cells, an increase in cellular excitability, and an increased risk of life-threatening cardiac arrhythmias. The *CACNA1C* gene is located on the short arm of chromosome 12 (12p13.3) [26].

There are two molecular genetic variants of Timothy syndrome. The most common one is named the “classic variant”. It is caused by a mutation in exon 8a of the *CACNA1C* gene and is characterized by polymorphism of clinical manifestations associated with the expression of this gene site in various tissues of the body. The “atypical” variant is less common and caused by mutations in exon 8 of the *CACNA1C* gene, leading to more pronounced prolongation of the QT and QTc interval and ventricular arrhythmias, most of which are drug-induced or associated with the use of anesthesia [25]. The “atypical” variant of Timothy syndrome is characterized by the maximum expression of the *CACNA1C* gene in exon 8 in the heart and brain (80% *CACNA1C* mRNA) [25].

QT syndrome type V (SQT5) is another genetic heterogeneous disease associated with impaired functioning of Cav1.2 channels. It is characterized by a decrease in QT interval ≤ 300 ms and the appearance of a high symmetrical peak-shaped T wave. Mutations leading to a shortening of the AP, which are pathological for this syndrome, were found in the gene-encoding K^+^ channels and the *CaCNB2b* gene (locus 10p12.33) encoding the β2-subunit of Cav1.2 [27].

The *CACNB2b* gene encodes 660 amino acids of the β2-subunit Cav1.2 [28]. This gene is mainly expressed in heart cells. The influx of Ca^2+^ ions into the cytosol is reduced in the channels with defective β2-subunits, which leads to a decrease in the I_L,Ca_ current. Phenotypically, mutations in the *CACNB2b* gene can lead to not only shortening of the QT interval (SQT5) but also to Brugada syndrome (BS) or a combination of both [28].

Brugada syndrome is known as a form of cardiac channelopathy associated with a high risk of SCD. It is believed that BS accounts for 20% of mortality in young people and men aged 30–40 years without structural pathology of the myocardium. The diagnostic hallmark of the syndrome is distinctive changes in the electrocardiogram (ECG) in the form of right bundle branch block and elevation of the ST segment in the right pericardial leads (V1-V3) [17,29]. This pathology is associated with mutations of the *CACNA1C* and *CACNB2b* gene-encoding α- and β2-subunits of the Cav1.2 channel, respectively. In patients with mutations of these genes, the bandwidth of Cav1.2 is reduced, and the I_LCa_ current is decreased [27].

The role of Cav1.3 in rhythmogenesis and heart rate modulation has been established only recently [14]. Compared to other calcium channels, Cav1.3 channels are activated faster and at more hyperpolarized voltages, which is important in maintaining the pacemaking and regulation of heart automaticity [24].

In humans, the first channelopathy involving the *CACNA1D* gene-encoding Cav1.3 protein was identified in 2011 [14]. Initially, a loss-of-function mutation in an alternatively spliced exon was associated with congenital deafness. Further observations showed that patients with this pathology had bradycardia and SA node dysfunction (sinus node weakness syndrome) with preserved normal QRS and QT in ECG. In the experimental conditions on mice, the *CACNA1D* gene knockout was shown to lead to the appearance of viable and fertile offspring but with deafness, bradycardia, and the dysfunction of SA and AV nodes, resulting in pathological functional disorders close to their human counterparts [14]. Experimental data suggest that the C-terminus of the Cav1.3 protein can function as a transcription regulator in atrial CM and modulate the expression of the myosin II light chain and small conductance calcium-activated K^+^ channel [14].

In ventricular CM, L-type Cav channels are the main structures maintaining myocardial contractility. Genetic or posttranslational modifications of Cavα1- and Cavβ2-subunits can lead to significantly reduced left-ventricular contractility and the development of ventricular tachycardia [30]. Defects in these channels lead to an increase in the duration of their open state. As a result, the mechanism of Ca^2+^-induced Ca^2+^ release (CICR) from the sarcoplasmic reticulum (SR), which is the trigger for the onset of contraction, becomes disrupted. The L-type Cav channels’ activity in the ventricular CM are usually transient and beneficial, but chronic irritation can become pathological [31]. To restore the functions of the CM associated with the malfunction of Cav1.2 and Cav1.3 channels, a method of targeted mobilization of a pre-synthesized pool of subsarcolemmal Cav1.2 channel-containing vesicles/endosomes into the CM sarcolemma has been developed [31].

### 2.2. R-Type Ca^2+^ Channels (Cav2.3)

R-type Ca^2+^ channels (Cav2.3) are encoded by the *CACNA1E* gene and are intermediate in electrophysiological properties between L- and T-type Cav channels. These channels are found in the pacemaker cells of the brain and heart, where they provide repetitive firing of action potentials [32]. Among other representatives of calcium channels, pharmacoresistant R-type channels are considered the most “mysterious” [33]. Their structure has not been sufficiently studied. Most of them are encoded by the *CACNA1E* gene and are expressed as different Ca_v_2.3 splice variants (variant Ca_v_2.3a to Ca_v_2.3e or Ca_v_2.3f) as the ion conducting subunit [8]. In humans, an R-type channel has been found so far only in thalamic neurons but proposes a cardiac mechanism of action for this channel [34]. The basic information on these channels in cardiomyocytes was obtained in animal models and isolated cells [8,32,33,34,35,36]. Owning to research on experimental models, it is now clear that cardiac pharmacoresistant R-type Ca^2+^ channels play an important role in regulating heart rate, pacemaking, and cardiac conduction.

These channels are highly conserved, and it can be assumed that human homologs of R-channels have similar properties to animal channels. However, further studies are needed to perform a comparison of animal and human R-channel data.

Like other HVA Ca^2+^ channels, Cav2.3 channels form multi-subunit complexes consisting of pore-forming α1- and α2-subunits, one of several cytoplasmic β-subunits and extracellular δ-subunits. The Cavα_1_-subunit is a pseudotetrameric protein with four homologous repeats (I–IV), which consist of six spiral segments penetrating the membrane (S1–S6). Four segments (S1–S4) in each repetition form a voltage sensor module, while the remaining two segments (S5–S6) of all repeats make up the majority of the pore domain (PD) and activation gates. The pore and the selective filter possessing a conserved Ca^2+^-selectivity filter motif ([T/S] × [D/E] × W) are formed by re-enterable pore loops (p-loops) between segments S5 and S6, which partially re-enter the pore region. Three inter-domain linkers and phosphorylation sites in the cytoplasmic domain of the channel are involved in inactivation, the association of auxiliary Cavß-subunits, and intracellular modulation [8].

R-type Ca^2+^ currents have been known to be resistant to most subtype-specific organic and peptide Ca^2+^ channel blockers [37], but they are also blocked by Ni^2+^ and are sensitive to Zn^2+^ and pharmacological regulators of store-operated Ca^2+^ channels, in particular, to isoproterenol [8]. These data are important for further research in the field of studying the possible contribution of R-type Ca^2+^ channel dysfunctions to the development of chronic human heart diseases and the pursuit of promising blockers for their therapy.

So far, no inherited disease is known for the *CACNA1E* gene, but recently, spontaneous mutations leading to early death in humans were identified [34].

### 2.3. T-Type Ca^2+^ Channels

T-type Ca^2+^ channels (Cav3.1, Cav3.2) («transient», «short-term») (low threshold or fast transient) mean the opening time of the channel [2,3].

Unlike HVA, which opens at −20 mV, Cav3 channels have a low threshold since their activation happens near −40 mV (in SAN cells, the channel activation threshold is −55 mV) [18,21]. The Cav3 channels are characterized by high sensitivity to the blocking action of Ni^2+^, low sensitivity to dihydropyridines and amiloride, and low conductivity (~8 pSm). The activity of the channels is regulated by G-protein-coupled receptors (GPCRs), which are blocked by mibefradil, Ni^2+^ (especially Cav3.2), and curtoxin [3,19].

T-type calcium channels consist of a single pore-forming α1-subunit that has two key structural determinants of Cav channel gating as well as ion selectivity and permeability. The Cav3 pore-forming α1-subunit is a relatively large plasma membrane protein of about 260 kDa organized into four hydrophobic domains (DI–DIV), each of which consists of six transmembrane helices (S1–S6) (Figure 3).

Similar to L-type Ca^2+^ channels, the voltage-sensitive channel module (S4) is formed by positively charged rich arginine/lysine, while the selectivity and channel ion conductivity depend on the re-entrant linkers connecting S5 and S6 modules and forming a P-loop. The four TM modules are linked together by several intracellular loops connecting the S6 module of the upstream domain to the S1 module of the downstream domain, which in combination with the NH2- and COOH-termini, provide a site/center for channel regulation by various signaling molecules and other protein partners, including the βγ-dimer of the G-proteins, PKA, calcineurin, CaMKII, syntaxin−1A, stac1, CACHD1, spectrin α/β, ankyrin B, etc. Furthermore, T-type channels undergo several posttranslational modifications, such as phosphorylation, glycosylation, and ubiquitination, which contribute to the expression and activity of the channel [3,38].

T-type calcium channels exhibit variations in their electrophysiological and pharmacological properties that can be explained by the existence of several channel splice variants [39]. These variants include Cav3.1, Cav3.2, and Cav3.3, which are encoded by the genes *CACNA1G*, *CACNA1H*, and *CACNA1I*, respectively, in humans [21,39].

T-type channels are formed by numerous variants of α1-subunits, among which the Cav3.1 channels made of (α1G) subunits are found in the heart. These channels are expressed mainly in sinoatrial and atrioventricular node cells, where along with Cav1.3 channels (α1D), they play an important role in generating spontaneous excitation of pacemaker cells. T-type channelopathies that drastically impair cardiac automaticity are considered rare [40]. They are associated with severe hereditary diseases that lead to sudden cardiac death. One such disease is myotonic dystrophy type I (MD1), known as Steinert disease, associated with a DMPK gene defect and channelopathy caused by mutations in genes associated with cardiac function (i.e., *TNNT*, *TNNT2*, *TTN*, *TPM1*, *SYNE1*, *MTMR1*, *NEBL*, and *TPM1*), including *CACNA1A* and *CACNA1H* [41].

Patients with MD1 suffer from defects in conductivity and atrial or ventricular tachyarrhythmia. The disease progresses with aging and becomes complicated by second- and third-degree heart blockage and left ventricular hypertrophy [41]. Histopathological analysis of the affected hearts in patients with MD1 showed fibrosis and multifocal disintegration of myofibrils [41].

Cav3.1 channel dysfunction was detected in patients with sinus node weakness and heart blockage caused by congenital autoimmune disease of the cardiac conduction system [24].

In ventricular CM, the population of Cav3.1 and Cav3.2 channels is uncommon. Therefore, their role in the regulation of myocardial contractility is insignificant. Transient expression of T-type Ca^2+^ channels occurs in the embryonic heart [42]. In a mouse model, it was demonstrated that the Cav3.2 channels are predominantly expressed in the embryonic heart from 9.5 to 18 days of embryonic development. At the same time, Cav3.1 channels are also expressed, but their expression level is significantly lower than Cav3.2 [43]. The functional role of these channels in the embryonic heart remained unknown. The T-type Ca^2+^ channels were suggested to be involved not only in the regulation of cell proliferation of prenatal CM but may also be included in the processes of cell growth in the differentiated heart [43]. Indeed, it turned out that Cav3.1 and Cav3.2 channels can be re-expressed in ventricular myocytes with pathological hypertrophy and myocardial infarction [43,44]. However, the increased expression of Cav3.2 channels is limited to the myocardial lesion zone and has a regional and temporary nature [43]. The hypertrophic overloads of the heart from α1G-transgenic mice demonstrated that the mice were resistant to pressure overload, isoproterenol, and cardiac hypertrophy caused by physical exertion. These mice also had no cardiac pathology, despite a significant increase in the influx of Ca^2+^ into CM. Unlikely, α1G-/-mice showed enhanced hypertrophic reactions after cardiac overload by pressure or the infusion of isoproterenol. Pathological hypertrophy in α1G-/-transgenic mice was reversed using the α1G-transgene, which proves the importance of Cav3.1 channels in cardioprotection and the prevention of cell remodeling [43].

## 3. Store-Operated Calcium Ca^2+^ Channels

Ca^2+^ ions play an important role in many physiological processes, including pacemaking, contraction, the release of neurotransmitters, ECC, gene expression, etc. Unsurprisingly, with a relatively low [Ca^2+^]_i_, a significant amount of Ca^2+^ is preserved by cells in intracellular Ca^2+^ stores. In CM and vascular smooth muscle cells (VSMCs), Ca^2+^ is stored mainly in SR. Calcium levels in cells are precisely regulated by various transporters and ion channels. Reduction in the intra-organelle concentration of Ca^2+^ serves as a signal to refill the Ca^2+^ stores through a store-operated calcium entry mechanism (SOCE).

### 3.1. Cardiac Ryanodine Receptor (RyR2)

Ryanodine receptors (RyR2s) are Ca^2+^-permeable ion channels in the membrane of the SR. These channels are responsible for local Ca^2+^-induced Ca^2+^ release from the SR. The Ca^2+^ released (Ca^2+^ sparks) activates contraction, pacemaking, and ECC, and affects other Ca^2+^-dependent intracellular processes.

RyRs are found in both atrial and ventricular CMs, as well as in vascular smooth muscle cells [45].

Currently, three isoforms of the RyRs have been characterized, one of which is the type 2 ryanodine receptor (RyR2), which has been found in CM. Modern innovations in cryo-electron microscopy have made it possible to obtain a number of near-atomic RyR structures that have contributed to a better understanding of the RyR architecture [46]. The RyR2 consists of four subunits combined into a homotetramer and the FC-binding protein calstabin 2 in a stoichiometric ratio of 1:4 [47].

The distinctive feature of the channels formed by RyR2 is that they are activated by an extracellular influx of Ca^2+^ as a result of the Cav1-RyR2 interaction that triggers local Ca^2+^ release [46].

The structure of these channels, as well as the molecular mechanisms responsible for cardiac pathology induced by the dysfunction of RyR2 and other cardiac Ca channels, were considered in detail in fundamental review papers published earlier [48,49]. The RyR2 macromolecular complex is one of the most well-studied multiprotein complexes [49]. A diverse array of RyR2-interacting proteins directly regulates RyR2 channel activity by binding to the pore subunit (e.g., FK506-binding protein−12.6 (FKBP12.6), calmodulin (CaM), calsequestrin-2 (CASQ2), junctin (JCTN), triadin (TRDN), and βII-spectrin [49]. CASQ2 binds to RyR2 via both JCTN and TRDN. RyR2 is strongly regulated by luminal Ca^2+^ levels, either by direct Ca^2+^ binding to RyR2 or by luminal Ca^2+^ interacting with CASQ2, JCTN, and TRDN [50]. Other proteins in the RyR2 macromolecular complex regulate the level of RyR2 posttranslational modification [49].

RyR2s are part of the pacemaker molecular mechanism that ensures heart automaticity. The contribution of the RyR2-dependent Ca^2+^ releases (Ca^2+^ sparks) to the automaticity is currently interpreted within the framework of a model dubbed the “calcium clock” [51]. According to the coupled-clock pacemaker cell system concept, the “clock” in pacemaker cells forms two competing oscillators: the “Ca^2+^-clock” mechanism based on the spontaneous release of Ca^2+^ from the SR and the “membrane clock” (“M-clock”), which includes surface membrane cation channels to ignite an AP [51]. The phenomenon of Ca^2+^ oscillations, underlying the “Ca^2+^-clock”, is associated with the local diastolic intracellular Ca^2+^ releases (LCRs), and it is independent of the membrane potential (MP). Indeed, spontaneous LCRs can be observed in the absence of changes in the MP and are a distinctive feature of the “Ca^2+^-clock” [52]. RyRs act as a gear in a “Ca^2+^-clock”, inducing rhythmical discharges of LCRs, which, in turn, activate an inward current (I_NCX1_) that prompts the “M-clock” to start an AP [51]. This electrogenic transport mechanism generates an internal ion current, which contributes to the onset of diastolic depolarization. In the final phase of diastolic depolarization, Cav channels cooperate with a Na^+^/Ca^2+^-exchanger (NCX) to raise the MP to the threshold value of the AP [51]. During the AP phase, the intake of Ca^2+^ through LTCC channels refills the leakage of Ca^2+^ from the SR, allowing new diastolic intracellular Ca^2+^ release to occur in the next cycle. Consequently, under normal physiological conditions, the “Ca^2+^-clock” and the “M-clock” synchronize the pacemaker activity of the heart cells and create a reliable basis for heart automaticity [51,53].

Considering the importance of the RYR2 in providing heart automaticity, it seems quite obvious that mutations of genes encoding these receptors or proteins interacting with RYR2 in the CICR would lead to catastrophic consequences for the organism [54]. Indeed, studies using a mouse model have shown that a RYR2 mutation with a locus in the CaM-binding site (reducing receptor inactivation) causes cardiac hypertrophy, heart failure, and early sudden death [55,56].

Clinical studies suggest that enhancing the interaction of CaM-RyR2 may represent an effective therapeutic strategy for the treatment of cardiac arrhythmias and heart failure. In support of this idea, the mutation of GOF CaM-M37Q and the reinforcement of the CaM-RyR2 interaction have been demonstrated to be able to suppress the spontaneous release of Ca^2+^ from the SR and catecholaminergic polymorphic ventricular tachycardia [57].

In chronic heart diseases accompanied by cardiac arrhythmias, there is an increase in the activity of Cav1.2 channels, which leads to an increase in their permeability to Ca^2+^ and, as a result, to CM calcium overload. At the same time, spontaneous Ca^2+^ releases induced by more frequent RYR2 openings form “pathological” calcium waves that are abolished and removed under physiological conditions with the participation of NCX1 and other molecular determinants of Ca^2+^ homeostasis (CASQ2, FKBP12, SERCA2a, etc.) [46]. It was found that NCX1 dysfunction and changes in its expression profile during arrhythmia lead to changes in atrial cell morphology and calcium handling together with dramatic alterations in the function of SAN [58,59].

It should be noted that the occurrence of pathological Ca^2+^ waves or an increase in the “threshold” of waves may be caused not only by the dysfunction of the cardiac calcium channels, in particular RyR2, but also by other pathogenetic factors, such as hypoxia and acidosis. Ca^2+^ alternans in ischemia can be taken as the arrhythmic triggers, leading to afterdepolarization and the substrate facilitating re-entry by inducing electrical alternans. More information about the cellular mechanism of cardiac alternans is found in the other studies [60].

Recent studies have shown that spontaneous arrhythmogenic “calcium waves” can result from genetic mutations of RyR2 but are more often due to an increase in the time of its open state [61,62,63,64,65]. Defects in its modular coupling with regulatory proteins of the cytosol, such as CaM, Epacl, PDE, FKBP12.6, PKA, PP1, calstabin, etc., or with Ca^2+^-binding proteins localized in the lumen of the SR (junctin, triadin, and calsequestrin), form temporary macromolecular complexes and can lead to a disruption of the RyR2 gate function [62,66,67,68,69]. Mechanisms of interaction of the partner molecules with RyR2 are built on a structural basis, while regulatory proteins (predominantly kinases) use RyR2 as a scaffold protein to form functional signal complexes that can modify a large number of other Ca^2+^-dependent molecules involved in the cascade signal transmission. The structure of these RyR2 multi-domain complexes and the mechanisms of regulation of their activity are still far from being fully understood [63,69].

It is noted that point mutations of RyR2-associated proteins or changes in their expression can dramatically affect the development of cardiac arrhythmias [70]. In particular, the expression level of serine/threonine protein phosphatases plays an important role in the pathogenesis of arrhythmias [71]. Serine/threonine protein phosphatases (PP1, PP2A, and PP2B) control the dephosphorylation of numerous cardiac proteins, including a variety of ion channels (Cav1.2, NKA, NCX, etc.), calcium-handling proteins (SERCA, junctin, and PLB), contractile proteins, MLC2, TnI, and MyBP-C [71,72], thereby providing posttranslational regulation of ECC and other heart functions. Accordingly, the dysfunction of this regulation can contribute to the development of cardiac arrhythmias. Atrial fibrillation (AF) is the most common heart rhythm disorder, and it is characterized by electrical and structural cardiac remodeling that among other factors includes changes in the phosphorylation status of a wide range of proteins, such as RyR2 [71]. It was found that a decrease in the concentration of PP1 caused by an increase in the level of PP1 regulatory proteins like inhibitor-1 (I-1), inhibitor-2, or heat-shock protein 20 in the sarcoplasm of ventricular CM leads to the rapid development of tachycardia and could cause sudden death [63,72], although an experimental increase in the concentration of PP1 in sarcoplasm prevented the development of arrhythmia, which was proven in experiments on mice overexpressing Ang II [73]. Mice characterized with a highly phosphorylated RyR2-S2808 site (S2808A+/+) demonstrated an increased sensitivity of RyR2 to Ca^2+^ during the dephosphorylation of PP-1 [74].

CaM kinase II is another enzyme that plays an important pathogenetic role in diseases accompanied by cardiac arrhythmia [52,54,75]. The effect of CaM kinase II on RyR2 is controversial. In pharmacological experiments using the method of embedding proteins in an artificial bilayer, some authors revealed the activating effect of this kinase on RyR2. On the other hand, others demonstrated its inhibitory effect [76]. Using molecular genetic methods (transgenic overexpression), more recent studies have found that CaM kinase II binds RyR2 and causes its phosphorylation at serine 2814 (RYR2-S2814). This, in turn, increases the frequency of Ca^2+^ spikes and the spontaneous local diastolic subsarcolemmal Ca^2+^ releases in the process of ECC [52]. It was noted that an increase in the level of phosphorylation of RYR-S2814 in SAN pacemaker cells led to the alteration of the “Ca^2+^-clock” regulation and the development of heart failure [77].

The inhibition of CaM kinase II by a specific blocker KN93 reduced the release of Ca^2+^ from the SR and slowed the heart rate [52]. In vitro studies have shown the possibility of using this blocker to relieve ventricular tachycardia caused by oxidative stress, which opens up prospects for the therapeutic use of KN-93 and its functional analogs in the treatment of arrhythmias [52].

In addition, an increase in the basal level of CaMKII through the phosphorylation of histone deacetylases (HDACs) activates myocyte-enhancer factor 2 (MEF2), which initiates the CaMKII/MEF-2 signaling pathway and hypertrophic remodeling of ventricular myocytes [78]. Among endogenous biologically active molecules that have arrhythmogenic effects, the most studied are neurohormones, such as endothelin-1 (ET-1), epinephrine, norepinephrine, etc., which act through Gαq-associated GPCRs and cAMP-dependent protein kinase A (PKA) [79,80,81]. Increased hormonal stimulation of these receptors leads to the hyperphosphorylation of RyR2 and the dissociation of calstabin 2 (FKBP12.6) from it. RyR2, deprived of this protein, loses its locking function, which leads to an increase in the time of its open state, a leakage of Ca^2+^, an increase in [Ca^2+^]i, and afterdepolarization, which can cause “fatal arrhythmia”, heart attack, and SCD [82]. To date, two genetic diseases associated with mutations in ventricular RyR2 have been described: catecholaminergic polymorphic ventricular tachycardia (CPVT), or familial polymorphic ventricular tachycardia, and arrhythmogenic right ventricular cardiomyopathy/dysplasia (ARVC/D) type 2 [83]. In patients with CPVT, the affinity of calstabin 2 to RyR2 is reduced due to a defect in RyR2 at its binding site to calstabin [82]. The use of molecular approaches in the strategy of the targeted therapy of CVD resulted in new drugs that suppress the hyperphosphorylation of p-RyR2 (Ser2808) and p-RyR2 (Ser2814), thereby stabilizing RyR2 and normalizing heart rate and the contractility of ventricular myocytes [84]. The ClinVar database (https://www.ncbi.nlm.nih.gov/clinvar/, accessed on 19 October 2023) describes 165 variants of pathogenic mutations of the *RYR2* gene. The most common mutations associated with CPVT are Ser2246Leu, Arg2474Ser, Asn4104Lys, Arg4497Cys, Pro2328Ser, 1.1-KB DEL, and EX3 [85]. Most of these mutations lead to amino acid substitution or the appearance of a premature stop codon and a disruption of the formation of a functional protein [86].

In addition to the *RYR2* gene, the encoding ryanodine receptor calcium release channel and mutations in five gene-encoding proteins from the SR calcium-release complex are involved in the pathogenesis of CPVT: CASQ2 (encoding cardiac calsequestrin), TRDN (encoding triadin), CALM1, CALM2, and CALM3 (encoding identical protein calmodulin) [87].

The development of arrhythmogenic dysplasia of the right ventricle is also associated with mutations in the *RYR2* gene. The ARVD2 locus was mapped to chromosome 1q42–q43 [88]. This disease is characterized by partial fatty or fibrous degeneration of the myocardium of the right ventricle, electrical instability, and sudden death [88]. The detection of *RyR2* mutations causing CPVT and ARVD2 opens the way to the pre-symptomatic detection of carriers of the disease in childhood, enabling early monitoring and treatment [87,88].

### 3.2. Ion Channels with Transient Receptor Potential (TRPC, TRPM7, TRPA1)

Transient receptor potential channels are nonselective cation channels of the TRP channel superfamily, uniting the related receptor proteins capable of being activated by the potential originating from the binding of the ligand to the receptor [89]. This superfamily is divided into a family of canonical TRP channels (TRPC) and several families whose names come from the name of the receptor, binding to which initiates the potential. Most TRPs are polymodal channels, so-called «coincidence detectors» that are activated by both physical and chemical stimuli [89]. TRP channels vary in degrees of selectivity and permeability to ions. TRPV1–TRPA1 and TRPM6/7 channels are more selective for Ca^2+^ ions [90]. In CM, they are localized in the sarcolemma adjacent mainly to intercalated disks and are activated by phospholipase C (PLC) via Gaq-associated G-protein-coupled receptors [89].

The common structural features of these channels are four N-terminal ankyrin repeats, six short TM domains, and a pore-forming region localized between transmembrane domains 5 and 6. Like the other previously described channels, TRP channels with partner proteins and kinases form signaling complexes that can be involved in the pathogenesis of various cardiovascular diseases [91,92,93,94].

The existing data suggest that several types of TRP channels (TRPC3, TRPC6, TRPV1, TRPV3, TRPV4, TRPA1, TRPM6, and TRPM7) may play a central role in the progression of fibroproliferative disorders in the heart and blood vessels and contribute to both acute and chronic inflammatory processes involved in them [91,95].

The family of canonical TRP channels consists of proteins closely related to the *Drosophila* channel proteins of the same name involved in photoreception [96]. This family includes seven subfamilies (TRPC1–TRPC7), of which proteins of the TRPC1, -3, -4, -6, and -7 subfamilies were found in CM [97]. To date, the greatest interest is focused on mechanosensitive TRPC channels. In case of their molecular «breakdown», these channels begin to pass an increased flow of Ca^2+^ ions, and, thereby, activate processes involving the pathological remodeling of CM [91,97,98]. In addition, TRPC7 also mediates apoptosis, thereby contributing to the process of heart failure [99].

All TRPC channels are dependent on receptors associated with PLC since they are directly or indirectly activated by phospholipid products formed due to activation of this enzyme and the induction of the hydrolysis of membrane phospholipids. TRPV1, -2, and -5 channels are activated by binding IP3 to the receptors and are responsible for SOCE. In this case, the interaction of TRPC with Orai protein and stromal interacting molecule 1 (STIM1) is noted [100]. Coupling between the Ca^2+^ entry and the intracellular Ca^2+^ stored in CM and VSMCs is mainly mediated by stromal interaction molecule, STIM1, located in the endoplasmic reticulum and ORAI1 membrane protein. These proteins are the primary components of the calcium-release activated calcium (CRAC) channel (Figure 4) [100]. In response to a decrease in the concentration of Ca^2+^ in SR, STIM1 is homooligomerized and translocated to the SR-PM contact sites, where it colocalizes with Ca^2+^-ATPase (SERCA), IP3R, and Orai1, forming the Ca^2+^ selective Orai1 pore. However, it is still unclear how such a close arrangement of proteins is conducted [100].

The mechanism of Ca^2+^ store filling with the participation of these proteins is as follows. After Ca^2+^ store depletion, the STIM1 protein located in the ER undergoes a complex conformational rearrangement, which results in STIM1 translocation into discrete ER plasma membrane junctions, where it directly interacts with the plasma membrane protein Orai1. Orai 1 triggers the recruitment of TRPC1 into the plasma membrane where it is activated by STIM1. TRPC1 and Orai1 form discrete STIM1-gated channels for the entry of Ca^2+^ into the lumen of the ER [101,102]. In addition, STIM1 can also activate TRPC1 through its C-terminal polybasic domain and is distinct from its Orai1-activating domain, SOAR [101].

TRPM2 is the second member of the TRPM subfamily that includes eight members, specifically TRPM1–8. TRPM2 is widely expressed in CM, where it forms a Ca^2+^-permeable cation channel and serves as a cellular sensor for oxidative stress or inflammatory response [103,104,105].

The N-terminus is composed of four melastatin homology regions, homology region pre-S1 (melastatin homology regions (MHRs), and homology regions (HRs). The channel domain contains six TMs (S1–S6), corresponding to a voltage-sensor-like domain; the pore is formed by the loop between S5 and S6. The C-terminus is composed of TRP and the coiled-coil domain (CC) (Figure 5).

TRPM2 channels are activated by ADPR, Ca^2+^, H2O2, and other reactive oxygen species (ROS). They serve as cellular sensors for oxidative stress, mediating oxidative stress-induced [Ca^2+^]i increase and contributing to pathological processes in many cell types, including CM. The overexpression of Trpm2 induces cell injury and death by Ca^2+^ overload or enhanced inflammatory response [103,105].

Mutations in genes encoding closely related TRPM4 channels lead to impaired automatism, conduction, and the appearance of hereditary progressive familial heart block type I (PFHBI) [106,107]. It is also assumed that some forms of provoked cardiac arrhythmia may occur due to a single gain-of-function mutation in TRPM4. To date, 47 mutations of the TRPM4 channel have been registered in the Human Gene Mutation Database [108,109].

TRPM7 channel mutations are especially dangerous in the prenatal period, as they can lead to intrauterine fatal arrhythmia and fetal death or a change in the myocardial transcription profile in adulthood and the deterioration of ventricular contractile function, conduction, and repolarization [110,111].

There is evidence of a wide involvement of TRP channels in the pathogenesis of CVD caused by hypoxia and oxidative stress, as well as ischemia-reperfusion (I/R) [90,104,112,113,114].

It is known that during hypoxia and I/R, there is a formation of ROS and an accumulation of lipid peroxidation products, including unsaturated aldehydes, such as acrolein and 4-hydroxynonenal, which are TRPA1 agonists, in cardiac tissue [115]. The mechanism of the toxic action of unsaturated aldehydes on CM is associated with their high electrophilicity and the ability to covalently bind to cysteine residues in the TRPA1 molecule, leading to the opening of these channels and an increase in sarcoplasmic reticulum Ca^2+^ release flux into the cytosol [116]. The overexpression of TRPC1 channels also contributes to Ca^2+^ leakage from the SR [117]. As a result, the Ca^2+^ overload of CM leads to impaired contractility, heart failure, and myocardial infarction [117].

The pathogenic significance of TRP channels in the development of heart failure, coronary artery disease, arterial and pulmonary hypertension, coronary microvascular dysfunction, and atherosclerosis is evident [91,93,94,118,119,120,121,122].

TRPA1, TRPV1–4, and TRPC1–6 are expressed on the surface of endothelial cells and ensure the passage of Ca^2+^ ions into cells, regulating endothelial-dependent vasodilation in response to a number of signaling molecules, such as endothelial-derived hyperpolarizing factor (EDHF), NO, and prostacyclin [90,113,119,123]. In mice with TRPV1 and TRPC3 channel knockout, there was a decrease in aortic vasodilation in response to carbachol, which proves the involvement of these channels in endothelium-dependent vasodilation [119,124]. In experiments on three animal models (dog, rat, and mouse), it was demonstrated that the i.v. administration of the TRPV4 agonist, GSK1016790A, stimulated endothelial-derived EDHF-dependent vasodilation and led to a subsequent decrease in blood pressure [125]. The TRPV4 channels have been noted to participate in the coupling of endothelial-dependent vasodilation and the relaxation of VSMCs through interaction with RyR2 and the big conduction BKCa channel [126]. TRPV4/7, TRPC1/5/6/7, and TRPV1/2/4 are expressed in VSMCs and participate in myogenic regulation of the vascular tone [119,120,127]. Electrophysiological experiments demonstrated that the TRPM4 knockout mice had an increased vascular tone and developed hypertension [90,128], while the inhibition of TRPC6 reduced VSMC contraction [127]. TRPV1 channels are involved in the progression of the atherosclerotic process in the apolipoprotein E gene knockout mice [129]. Further studies have shown that other mechanosensitive TRP channels may also play a role in the development of atherosclerosis and coronary heart disease [130,131]. In addition, a disruption of the TRP channel expression or function may explain the observed increased cardiovascular risk in patients with metabolic syndrome [132].

Thus, TRP channels have broad cardiovascular plasticity. TRPC3, TRPC5, TRPC6, TRPV1, and TRPM7 are involved in the vasoconstriction and regulation of blood pressure and can be considered potential therapeutic targets for the treatment of chronic CVD, including cardiometabolic diseases, and myocardial atrophy [91,119,121,132,133,134].

## 4. Hyperpolarization-Activated Cyclic Nucleotide-Gated (HCN) Channels

HCN channels are nonselective cationic channels that are involved in the generation of pacemaker activity in heart and brain cells [135,136]. They belong to the family of channels operated by cyclic nucleotides, which are part of the superfamily of potential-operated potassium channels, the regulation of which is under the control of the autonomic nervous system [137]. There are four isoforms of HCN channels (HCN1–4) encoded by the same genes [138]. The function of HCN channels is to generate a the “funny current” (I_f_) or “pacemaker current” (I_f_) during the hyperpolarization phase of the AP [139,140].

HCN channels are expressed differentially. HCN4 is the dominant form found in the SAN. HCN2 is found in the His–Purkinje system. HCN1 is also expressed in the SAN, but it is less optimal for pacemaker targeting [139]. HCN1 and HCN2 transcripts are predominantly found in ventricular CM [140,141].

HCN channels are primarily selective to Na^+^ and K^+^, but their functioning directly affects the influx of Ca^2+^ ions into CM and has a regulatory effect on diastolic depolarization under physiological conditions [142,143,144,145]. Electrophysiological experiments using the patch-clamp technique revealed that the influx of Ca^2+^ through HCN2 channels is enhanced by increasing the time of their open state with an increase in the concentration of cAMP and is inhibited by the specific I_f_-blocker ivabradine [145].

HCN channels are tetrameric cation channels [139,146]. They consist of four subunits (Figure 6), which can be either the same or different from each other [139]. However, in vivo channels consisting of subunits of the same type are more common [146].

Additionally, HCN1 contains a di-arginine ER retention signal in the intrinsically disordered region of the C-terminus of HCN1 [147]. This signal controls the trafficking of HCN1 and negatively regulates the surface expression of HCN1 [146,147]. Deletion of the entire N terminus (residues 1–185) also prevented surface expression of HCN2 [148].

HCN isoforms (1–4) are highly conserved relative to TM and the binding site for cyclic nucleotides (80–90% identity) but have differences in activation and reactivation kinetics, depending on voltage and cAMP modulation [149,150]. For example, HCN4 exhibits the slowest activation and reactivation kinetics and opens at more negative potentials than other isoforms [151,152]. On the contrary, HCN1 demonstrates the fastest kinetics and opens at more positive potentials [152]. HCN4 is the most sensitive to cAMP, while the HCN1 subtype is faintly affected by cAMP and other cyclic nucleotides [139,153,154].

Cardiac activity is under hormonal control and modulated by mechanisms mediated by small proteins, β-adrenoreceptors, and cAMP [155,156]. An uncontrolled increase in the concentration of cAMP (sympathicotonia), or other non-canonical cyclic nucleotides, an increase in membrane expression, and the activation rate of HCN channels in the SAN can lead to modulation of the cAMP-dependent activation profiles and an increase in heart rate and SCD [155,157].

HCN channel dysfunction, decreased levels of connexin 40, connexin 43, and myocyte-enhancer factor-2C, and components of gap junction were noted in the hearts of transgenic mice with an overexpression of calreticulin [158]. The complex of these disorders underlies the development of arrhythmia, dilated cardiomyopathy, and SCD [158]. Age-related conduction disorders are also associated with HCN channel dysfunction [159].

Recent studies have shown the existence of four HCN channelopathies associated with different types of arrhythmia [153]. The mutation in exon 5 of the *HCN4* gene is functionally associated with a truncated protein that is unable to bind cAMP and, therefore, has a dominant negative effect on channel function. Missense mutations of the same gene have been found in families of patients with SAN dysfunction. The presence of these mutations led to recurrent syncopes, severe bradycardia (39 beats per minute), prolonged QT intervals, and polymorphic ventricular tachycardia [153]. The dysfunction of HCN channels, mainly HCN4, is associated with sick sinus syndrome and other arrhythmias, such as atrial fibrillation, ventricular tachycardia, and atrioventricular block. In recent years, several data have also shown that dysfunctional HCN channels (HCN1, HCN2, and HCN4) may play an important role in the pathogenesis of epilepsy [160]. Myocardial involvement is frequent in patients affected by neuromuscular disorders and is the main cause of death in some conditions (Table 1).

Currently, there are ongoing studies on the directed transport of recombinant genes directly into the heart [141]. In experiments on mice, adenoviral constructs expressing the HCN2 gene were delivered by epicardial injection into the root of the appendage of the left atrial appendage. Four days after the targeted delivery of the *HCN2* gene, spontaneous beats occurring at the injection site were detected. The heart rate in experimental mice was under the control of the autonomic nervous system, which was proved by stimulation of the heart rate with catecholamines and a decrease in heart rate because of stimulation of the left vagus nerve. Cells localized at the injection site showed an increased expression of *HCN2* channels and increased I_f_ currents [161].

These studies show the effectiveness of a targeted approach in the treatment of arrhythmias and congenital cardiac HCN channelopathies in humans [141,160].

However, for the successful gene therapy of HCN channelopathies in humans, additional mutant or chimeric HCN channel constructs will be required, which will have more positive activation and increased reactivity and will optimize the method of delivery of the therapeutic genes to the target cells of the heart. Research in this direction is underway [141,160].

## 5. Prospects of Using the Achievements of Molecular Biology in the Treatment of Chronic Heart Diseases

The cornerstone of the strategy of targeted molecular therapy is the idea of using drugs that act on subcellular structures involved in the mechanism of disease development. Currently, such drugs are mainly used for the treatment of certain types of cancer. In the therapy of cardiac diseases, the use of substances targeting certain molecular substrates has not yet become common. The most well-known drugs that can be attributed to this drug group are calcium channel blockers (CCBs). They have been used since the 1970s and have proven to be effective and reliable means in the treatment of CVD accompanied by rhythm disorders and decreased myocardial contractility. The broad use of CCBs in clinical practice was facilitated by their high anti-ischemic and antianginal efficacy and good tolerability, which were established during large clinical studies [162]. Over the past years, more than one generation of drugs of this group has changed. Modern CCBs (amlodipine, lacidipine, etc.) are substances that differ from their predecessors (verapamil, nifedipine, diltiazem, etc.) by the prolongation of action and a higher safety profile [163,164]. Drugs of this group have a cardioprotective effect by improving myocardial perfusion, reducing myocardial oxygen demand, and reducing the formation of free radicals and mitochondrial Ca^2+^ overloading of CM [163,164,165]. The disadvantages of CCBs as drugs are their rather wide range of action and low selectivity with respect to their molecular targets directly in the myocardium. Moreover, the use of these drugs is associated with an increased risk of developing proarrhythmia and systemic toxicity, an increase in the defibrillation threshold, and, in some cases, an increase in mortality [166]. Such significant disadvantages of CCBs make it necessary to search for novel alternative drugs, the targets of therapeutic action of which are not the channel proteins themselves but the molecules modulating their activity. Such proteins can be fully attributed to CaM kinase II, which is, on the one hand, a key modulating enzyme of Ca^2+^ metabolism in chronic heart pathology, and on the other hand, a widespread protein found in other vertebrates, making it possible to study the effect of CaM kinase II blockers initially in vivo experiments and in situ [167,168].

It is obvious that other regulatory proteins that exert their cardiotropic effect through interaction with Ca^2+^ channels or their modulators can become targets for pharmacological strategy in the treatment of chronic heart diseases. We mentioned some of them previously [84,85]. Currently, the most promising are studies aimed at finding blockers of the NCX1 [169], various isoforms of phosphodiesterases [170], and subunits of voltage-operated channels of various types, including nonselective HCN1–4 and TRP channels [171,172]. The use of these “molecular tools” to influence the mechanisms of Ca^2+^ signaling in heart failure and other chronic heart diseases is the first step toward making new drugs whose therapeutic targets are small fragments of specialized molecules or their specific isoforms. Recent progress in the studies of the structure and properties of Ca^2+^ channels and Ca^2+^ handling proteins, as well as modern technologies using targeted nanoparticles and targeted gene delivery directly to the heart, allow us to hope that significant progress will be made in the treatment of severe CVD associated with heart rate and contractility disorders in the near future [141,173].

## 6. Conclusions

Most chronic heart diseases are characterized by disorders of heart rhythm, conduction, and contractility. In addition, a violation of the electrical activity of the heart, the appearance of extensive ectopic foci, and heart failure are all symptoms of a number of severe hereditary diseases. The molecular mechanisms leading to the development of heart diseases are associated with impaired permeability and excitability of cell membranes and are mainly caused by the dysfunction of cardiac Ca^2+^ channels. More than 100 varieties of ion channels have been found in the cells of the cardiovascular system. The relationship between the activity of these channels and cardiac pathology, as well as the general biological function of cardiac muscle cells, is intensively studied in experimental animal models in situ and in vivo. However, little is still known about the origin of genetic Ca^2+^ channelopathies in humans, the role of the dysfunctions of various types of Ca^2+^ channels in the phenomenon of cardiac alternans, and the development of cardiac pathology in humans. Most research in this area is conducted using genetically modified animal models or in vitro. To what extent are the data obtained correct in relation to people? This question is still difficult to answer. Some Ca^2+^ channels, for example, subpopulations of R-type Ca^2+^ channels, are found only in animals. Other channels (subpopulations of T-channels) arise in early ontogenesis and undergo involution during natal (perinatal) development. Do they matter in the pathophysiology of the heart? The role of nonselective (hyperpolarization-activated cyclic nucleotide-dependent channels (HCN) and transient receptor potential (TRP)) in the development of cardiac pathology requires clarification. There are still no answers to questions related to the specific treatment of Ca^2+^ channelopathies in humans. However, with the use of the latest innovative instrumental and molecular genetic research methods, new opportunities for solving these problems are emerging.

## Figures and Tables

**Figure 1 ijms-24-15682-f001:**
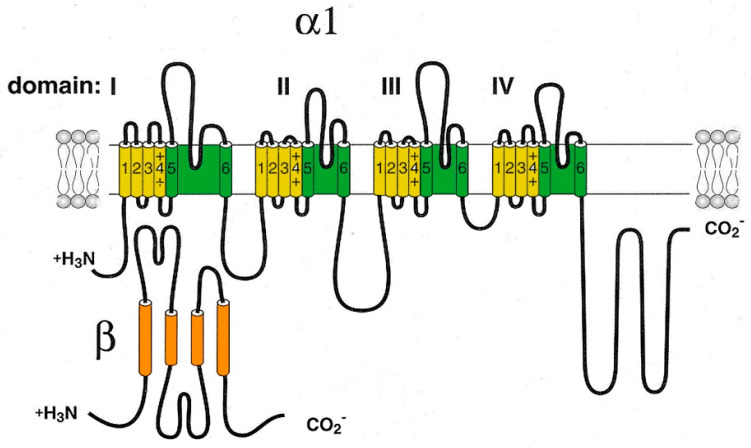
Schematic representation of the pore-forming channel α1-subunit Cav1. The α1-subunit includes four homologous domains (I–IV), each of which consists of six transmembrane segments (1–6). Segments 5 and 6 together with the linker peptide (linkers 5–6) form a selective pore permeable to Ca^2+^ ions. Segment 4 is a voltage-sensing module. The β-subunit is involved in the inactivation and closure of the channel. Both N- and C-termini are in the cytosol.

**Figure 2 ijms-24-15682-f002:**
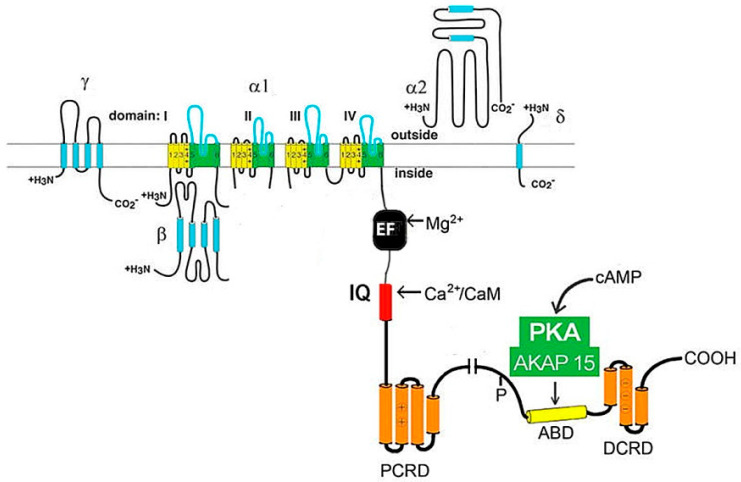
The cardiac Ca_V_1.2 channel signaling complex. ABD, AKAP15 binding domain; DCRD, distal C-terminal regulatory domain; PCRD, proximal C-terminal regulatory domain; scissors, site of proteolytic processing.

**Figure 3 ijms-24-15682-f003:**
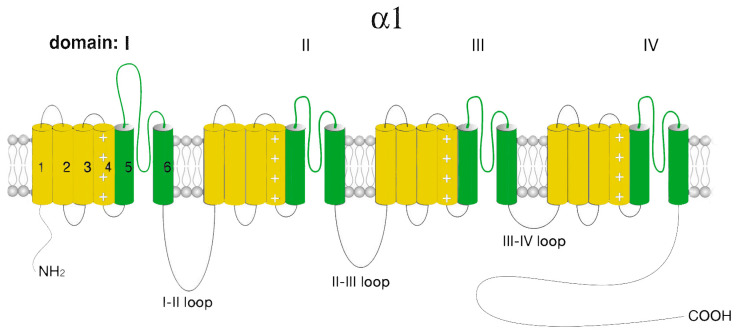
Schematic representation of the Cav3 α1-subunit. The α1-subunit includes four homologous hydrophobic domains (I–IV), each of which consists of six transmembrane segments (1–6). Segments 5 and 6 together with linkers 5–6 form a highly conserved pore loop permeable to Ca^2+^ ions. Segment 4 is a voltage-sensing module. The amino (NH_2_ and carboxyl (COOH) termini and the cytoplasmic interdomain l-II-, II-III-, III-IV loops are in the cytosol.

**Figure 4 ijms-24-15682-f004:**
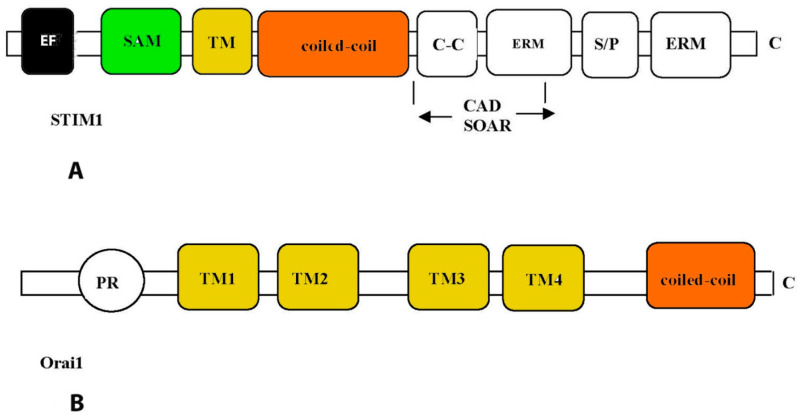
Structural features of STIM1 (**A**) and Orai1 (**B**) proteins. Functional domains of proteins are enclosed in rectangles: EF—Ca^2+^-binding motif “EF-hands”; SAM—so-called “sterile”-α-motif (sterile-α-motif); TM—transmembrane domain; ERM—protein binding domain of the ERM complex; S/P—domain enriched with serine and proline; C-C—domain enriched with lysine; CAD—domain responsible for channel activation; SOAR, STIM1-Orai—activation region; PR—domain enriched with proline and arginine; TM1–4—transmembrane domains.

**Figure 5 ijms-24-15682-f005:**
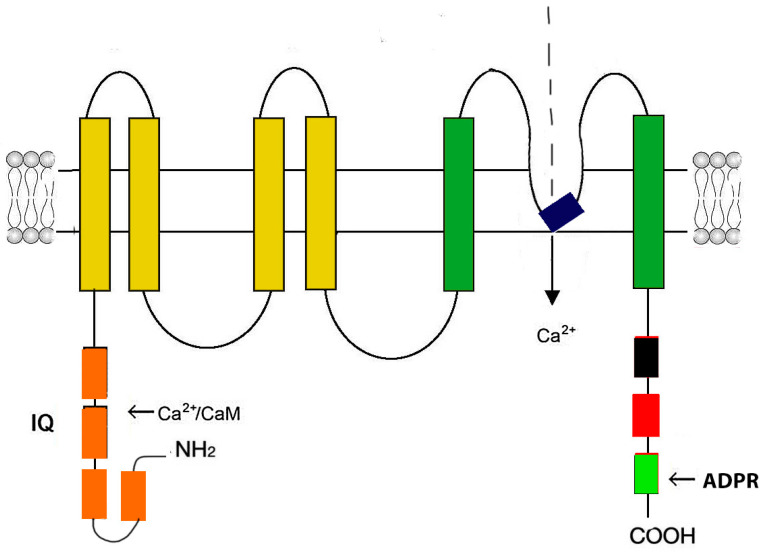
Schematic representation of the Trpm2 monomer structure. The Trpm2 monomer consists of six transmembrane domains (yellow and dark green rectangles). The pore-firming loop is located between segments 5 and 6 (black rectangle). Both N- and C-termini are in the cytosol. The N-terminus contains four modules of the Trpm subfamily melastatin homology domain (MHD) (orange rectangles). In the second MHD, there is an IQ-like motif that binds Ca^2+^-calmodulin. The C-terminus contains a Trp box (TRP) (black rectangle), a coiled-coil domain (CC) (red rectangle), and a adenosine diphosphate ribose (ADPR) pyrophosphatase homolog domain.

**Figure 6 ijms-24-15682-f006:**
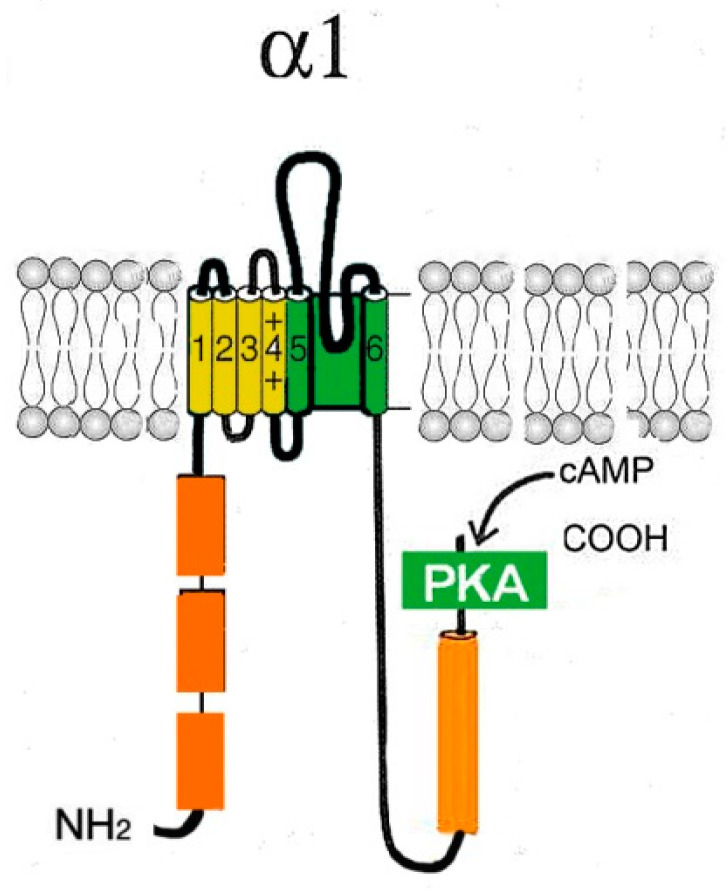
Schematic representation of the α1-subunit of the HCN channel. Each HCN monomer contains six TM segments (1–6), including a positively charged potential sensor (S4) and P-region located between the pore-forming segments 5 and 6. The C-terminus contains two conserved structured regions: the C-linker, contributing to tetramerization, and the CNBD, which allows for modulation by cAMP. A region at the end of the N-terminal domain (orange rectangles) is conserved among HCN channel subtypes and has been found to be important for channel trafficking.

**Table 1 ijms-24-15682-t001:** Ca^2+^ channels and associated channelopathies (based on OMIM, the Online Mendelian Inheritance in Man database).

Ca^2+^ Channel	Gene	Channelopathy, Syndromes	OMIM
**Cav**			
Cav1.1	*CACNA1S*	Hypokalemic periodic paralysis type 1 Normokalemic periodic paralysis Malignant hypothermia susceptibility 5	170400 170600 601887
Cav1.2	*CACNA1C*	Timothy syndrome Long QT syndrome 8 (LQT8) Brugada syndrome 3	601005 618447 611875
Cav1.3	*CACNA1D*	Sinoatrial node dysfunction and deafness syndrome	614896
		Primary aldosteronism, seizures, and neurologic abnormalities	615474
		Autism spectrum disorder (with or without more severe manifestations including intellectual disability, neurological abnormalities, primary aldosteronism, and/or congenital hyperinsulinism)	Not listed in OMIM
Cav1.4	*CACNA1F*	Aldosterone-producing adenomas Congenital stationary night blindness type 2 X-linked cone–rod dystrophy 3 Aland Island eye disease	Not listed in OMIM 300071 300476 300600
Cav2.1	*CACNA1A*	Familial and sporadic hemiplegic migraine type 1 with or without progressive cerebellar ataxia	141500
		Episodic ataxia type 2	108500
		Spinocerebellar ataxia type 6	183086
		Early infantile epileptic encephalopathy 42	617106
		Congenital ataxia	Not listed in OMIM
	*CACNA1B*	Neurodevelopmental disorder with seizures and non-epileptic hyperkinetic movements	618497
Cav2.3	*CACNA1E*	Early infantile epileptic encephalopathy 69	618285
Cav3.1	*CACNA1G*	Spinocerebellar ataxia type 42 Early-onset spinocerebellar ataxia type 42 with neurodevelopmental deficits (childhood-onset cerebellar atrophy)	616795 618087
Cav3.2	*CACNA1H*	Familial hyperaldosteronism type IV Aldosterone-producing adenomas	617027 Not listed in OMIM
Cav3.3	*CACNA1I*	Neurodevelopmental disorder with epilepsy and intellectual disability	Not listed in OMIM
**RyR**			
RyR2	*RYR2*	Arrhythmogenic right ventricular dysplasia/cardiomyopathy type 2 Stress-induced polymorphic ventricular tachycardia	600996
		(catecholaminergic polymorphic ventricular tachycardia 1)	604772
**TRP**			
TRPC3	*TRPC3*	Spinocerebellar ataxia	602345
TRPC6	*TRPC6*	Glomerulosclerosis, focal segmental, 2	603965
TRPV3	*TRPV3*	Olmsted syndrome	614594
TRPV4	*TRPV4*	Brachyolmia type 3	113500
		Digital arthropathy–brachydactyly, familial	606835
		Hereditary motor and sensory neuropathy, type IIc	606071
		Metatropic dysplasia	156530
		Parastremmatic dwarfism	168400
		Scapuloperoneal spinal muscular atrophy	181405
		SED, Maroteaux type	184095
		Spinal muscular atrophy, distal, congenital nonprogressive	600175
		Spondylometaphyseal dysplasia, Kozlowski type	184252
TRPM1	*TRPM1*	Night blindness, congenital stationary (complete), 1C, autosomal recessive	613216
TRPM4	*TRPM4*	Progressive familial heart block, type IB	604559
TRPM6	*TRPM6*	Hypomagnesemia 1, intestinal	602014
TRPA1	*TRPA1*	Episodic pain syndrome, familial	615040
TRPML1	*TRPML1*	Mucolipidosis IV	252650
PKD2 (TRPP1)	*PKD2*	Autosomal dominant polycystic kidney disease	613095
**HCN**			
HCN1	*HCN1*	Dravet syndrome	Not listed in OMIM

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
