# Peer review of "The Dysfunction of Ca^2+^ Channels in Hereditary and Chronic Human Heart Diseases and Experimental Animal Models"

_ijms, 2023, doi:10.3390/ijms242115682_

Round 1
Reviewer 1 Report
The author provides a comprehensive review on calcium channel dysfunctions responsible for chronic human heart diseases. The text is extensively documented by many references in the bibliography, and the review is well organized in chapters according to calcium channel families with helpful schemes illustrating protein structures. One criticism that could be made is that the information is dense and sometimes provided as if in a catalog, with little information on the molecular or electrophysiological mechanisms responsible for the cardiac pathology induced by channel dysfunction. Furthermore, there is very little mention of the pathophysiological controls leading to ectopic or anachronistic expression of specific channels, which in some cases can lead to heart failure or arrhythmias.
I have only a few specific points:
1) P.8, paragraph 6: the humoral modulation of ectopic (re)-expression of a1H (for example upon hyperaldosteronism) should also be discussed here as a possible additional cause of lethal arrhythmias.
2) P.9, para 1: calcium is not stored in the mitochondria, it passes into the mitochondrial matrix when calcium rises in the cytosol, and leaves the mitochondria when cytosolic levels come back to resting values.
3) P.9. para 1: Reduction in the intra-organelle concentration of calcium (not intracellular)
4) P.9-10. The story of STIM1, ORAI1 and CRAC channel should be presented in another chapter. Indeed, there is a mixing between Ca-release channels (RyR and IP3R) and store operated plasma membrane calcium channels (that open for refilling when cellular calcium stores are empty)
5) P.12, para 4: “such as endothelin-1 (ET-1), aldosterone, epinephrine, which act through Gαq-associated GPCRs and cAMP-dependent protein kinase A (PKA)”. Aldosterone does not act through a GPCR
6) Fig. 5. The names of the boxes (MDH, TRP, CC) should be shown on the figure
Minor points:
7) P.7, Figure 3 legend: cytoplasmic interdomain I-II (not II-II)
8) P.8, para1: including the bg-dimer of G proteins (not of GPCR)
9) P.8, para 3: CaV3.1 channels/currents correspond to a1G (not a1G and a1H), while CaV3.2 corresponds to a1H.
10) P.9 and elsewhere: pacemaking (not pacemakering)
11) Ref.57 is not mentioned within the text; idem for ref. 36
12) Fig.6, legend: a1 subunit of the HCN channel (not a subunit)
Author Response
Guided by the recommendation of a respected reviewer, we have supplemented the manuscript with a paragraph and references to works that provide information on the molecular and electrophysiological mechanisms responsible for cardiac pathology caused by channel dysfunction.
The structure of these channels, as well as the molecular mechanisms responsible for cardiac pathology induced by the dysfunction of RyR2 and other cardiac Ca channels, were considered in detail in fundamental review papers published earlier [48,49].
We thank the distinguished reviewer for his critical remark, guided by which we supplemented the manuscript with information about the pathophysiological mechanisms leading to ectopic activity and the appearance of Ca2+ alternans.
It should be noted that the occurrence of pathological Ca2+ waves or an increase in the "threshold" of waves may be caused not only by dysfunction of the cardiac calcium channels, in particular RyR2, but also by other pathogenetic factors such as hypoxia and acidosis. Ca2+ alternans in ischemia can be taken as the arrhythmic triggers leading to afterdepolarization and also as the substrate facilitating reentry by inducing electrical alternans. More information about cellular mechanism of cardiac alternans be found in the works [60, 61].
We also agree with the minor comments of the respected reviewer and have amended the text, marking the corrections in blue.
Please see the attachment 2

Reviewer 2 Report
The review written by Shemarova includes a comprehensive summary of function and dysfunction of Ca2+ channels in the human heart disease.
I personally think the review is scientifically loud and is overall well written.
Please find some comments below:
*the numbering of the lines is a bit misleading
Abstract
The authors mention that more than 100 varieties of ion channels have been described in cardiovascular cells. I suggest specifying which cells (not here but in the introduction section) to make the reader aware on the cardiovascular cells’ active players in the heart calcium homeostasis.
Introduction
I suggest the author to revise this section, possibly including a cartoon where all the types of calcium channels are depicted.
a. Page1, line 37 the term “condition” should be probably substituted by “activity”?
b. Page 1, from line 41. I suggest the author to rephrase this following section. Readers should get information on the calcium channels in the healthy heart and afterwards in the diseases heart. However, I do not encourage putting too many details on the disease part as this is extensively included in the following sections.
c. Page 2, line 51, reference missing.
d. Page 2, from line 53, this part could be moved to the appropriate following section.
e. Introduction should also include a sentence to present the structure and “aims” of the review. Please include that.
f. Additional information, such as incidence, prevalence and/or numbers related to its morbidities can help the reader to understand its actual impact on the population.
g. What is the impact that such a complex and multifactorial disease has on the health system?
Following sections on the types of calcium channels
I suggest the author to revise all sections having in mind a common structure, such as 1) description of the calcium channel type 2) description of the structure of the channel 3) description of the function 4) description of the channel in disease conditions.
Moreover, please avoid redundance information (example, page 6 line 22? “Most of them are encoded by the CACNA1E gene and are expressed..” this concept was already formulated in the first sentence of the section).
Table 1
Table 1 is of great importance. I encourage the author to place right after the introduction, as it can help the reader to link the genetics with a specific calcium channel.
Future perspectives
Section 5 could be renamed as “Future perspectives” and could include the info starting from page 18, line 68 (after table 1).
Finally, cellular calcium homeostasis also includes mitochondrial calcium. A short notice on that would be appreciated.
Dear author,
minor English changes are required. Please read it through, when possible use concise statements and avoid repetitions.
Author Response
Please see the attachment 2

Reviewer 3 Report
Overall, this is a nice review. The author systematically summarized and discussed the Calcium channels in the heart. I have no major comments but the following suggestions.
In this review, the examples linking channel dysfunctions to heart diseases are only on the membrane. However, the intracellular Calcium handling process around calcium stores is also critical in triggering cardiac arrhythmias, see https://doi.org/10.1155/2019/8237071 & https://link.springer.com/article/10.1631/jzus.B1300177. This aspect should be reflected in the manuscript.
good enough
Author Response
Plea
We have supplemented the manuscript with additional information in accordance with the recommendations of a respected reviewer and thank the respected reviewer for a good assessment of our work.
It should be noted that the occurrence of pathological Ca2+ waves or an increase in the "threshold" of waves may be caused not only by dysfunction of the cardiac calcium channels, in particular RyR2, but also by other pathogenetic factors such as hypoxia and acidosis. Ca2+ alternans in ischemia can be taken as the arrhythmic triggers leading to afterdepolarization and also as the substrate facilitating reentry by inducing electrical alternans. More information about cellular mechanism of cardiac alternans be found in the works [60, 61].
se see the attachment 2

Reviewer 4 Report
This is a review article focused on Ca2+ channels in chronic heart diseases. This reviewer believes that the present review article is well-crafted and would like to provide the following comments as described below.
Major comment:
1. It would be more appropriate to delineate the "aim of this review article" at the conclusion of the Introduction section and include a "conclusion" section to wrap up this review article following the future perspective section.
Minor comments:
2. In line 17, it is noted that this review article has a single author. Therefore, the appropriate pronoun to use is "I" rather than "we".
3. Several typographical errors appear to be present in the manuscript. For instance, there is inconsistency in the font of the word "are" in line 295, and the use of bold formatting in lines 456 and 461 may need clarification if it was not the author's intention.
Nothing special except typos as described in the comments for the author.
Author Response
с
In accordance with “Major comment” of a respected reviewer, we have supplemented the manuscript with the specified paragraph and section.
The purpose of this review is to highlight the little-studied aspects of the effect of Ca2+ channel dysfunction on the development of chronic and hereditary heart diseases.
- Conclusion
Most chronic heart diseases are characterized by disorders of heart rhythm, conduction and contractility. In addition, a violation of the electrical activity of the heart, the appearance of extensive ectopic foci and heart failure are all symptoms of a number of severe hereditary diseases. The molecular mechanisms leading to the development of heart diseases are associated with impaired permeability and excitability of cell membranes and are mainly caused by dysfunction of cardiac Ca2+ channels. More than 100 varieties of ion channels have been found in the cells of the cardiovascular system. The relationship between the activity of these channels and cardiac pathology, as well as the general biological function of cardiac muscle cells, is intensively studied in experimental animal models, in situ and in vivo. However, little is still known about the origin of genetic Ca2+ channelopathies in humans and the role of dysfunctions of various types of Ca2+ channel in the phenomenon of cardiac alternans and the development of cardiac pathology in humans. Most research in this area is conducted using genetically modified animal models or in vitro. To what extent are the data obtained correct in relation to people? This question is still difficult to answer. Some Ca2+ channels, for example, subpopulations of R-type Ca2+ channels, are found only in animals. Other channels (subpopulations of T-channels) arise in early ontogenesis and undergo involution during natal (perinatal) development. Do they matter in the pathophysiology of the heart? The role of nonselective (hyperpolarization-activated cyclic nucleotide-dependent channels (HCN) and transient receptor potential (TRP)) in the development of cardiac pathology requires clarification. There are still no answers to questions related to the specific treatment of Ca2+ channelopathies in humans. However, with the use of the latest innovative instrumental and molecular genetic research methods, new opportunities for solving these problems are emerging.
We also agree with the minor comments of the respected reviewer and have amended the text, marking the corrections in blue.

Round 2
Reviewer 4 Report
This reviewer has no further comment.